# Brain-inspired Representation Transfer via Invariant Input-driven Continuous Attractors in a modular RNN framework

## Abstract

Conventional end-to-end deep neural networks often degrade under domain shifts and require costly retraining when deployed in unpredictable, noisy environments. Inspired by biological neural computation, we propose a modular framework in which each module is a recurrent neural network pretrained using a simple, task-agnostic protocol to learn robust, transferable features. We show that low-dimensional, input-driven continuous attractor manifolds, embedded in a high-dimensional latent space, yield task-invariant representations that enable robust transfer and resilience to temporal perturbations. At deployment, only a lightweight adapter needs training, allowing rapid adaptation to new tasks. Validated on the Dynamic Vision Sensor (DVS) Gesture benchmark and a custom rehabilitation action recognition dataset we collected, our framework achieves accuracy competitive with state-of-the-art methods, especially in few-shot settings, while requiring an order of magnitude fewer parameters and minimal training. By integrating biologically inspired attractor dynamics with cortical-like modular composition, the framework provides a practical route to robust, continual adaptation in real-world information processing.

## 1 Introduction

Artificial Intelligence (AI) systems deployed in real-world contexts–such as robotics, autonomous vehicles, and wearable assistants–must operate reliably in open-ended, unpredictable environments. In practice, they routinely encounter sensor noise, adversarial perturbations, data imbalance, and non-stationarity (Xing et al., 2025). Achieving robust performance under such conditions demands rapid domain adaptation (Liu et al., 2021), especially when training data are scarce or environments shift over time. This raises three fundamental challenges: 1) transferability: how to build representations that generalize across environments; 2) robustness: how to remain stable under noise and corrupted inputs; 3) cross-task adaptability: how to rapidly learn new tasks with minimal data.

Conventional end-to-end deep learning architectures are poorly suited to these challenges of real-world deployment. They assume matched training and testing distributions (Roy et al., 2021), leaving their task-specific features brittle under distributional shifts. Biological cognition, by contrast, achieves remarkable adaptability and generalization from limited supervision (Barry et al., 2007), drawing on evolutionarily conserved neural circuitry that can be flexibly reused across many tasks (Anderson, 2010; 2016). A key mechanism is the emergence of collective neural dynamics that self-organize into attractors–low-dimensional, stable activity patterns that remain consistent across diverse input conditions (Khona & Fiete, 2022). Attractors can encode core motion variables such as direction and position, and have been proposed as reusable cognitive "symbols" for fundamental concepts or operations (Nam et al., 2023; Shadlen & Newsome, 2001). Their tolerance to noise and representational stability make them well-suited for modular computation and transfer across tasks (Burak & Fiete, 2009; Mathis, 2024). Yet it remains unclear how randomly recurrent neural networks (RNNs) can develop continuous attractors that serve as robust, transferable representations, and how such dynamics can be recomposed for rapid adaptation with minimal supervision.

To address this gap, we propose a brain-inspired modular framework that explicitly embeds continuous attractor manifolds (Manjunath et al., 2012) as transferable representations for domain adaptation. Drawing on insights from neuroscience and dynamical systems (Hocker et al., 2025), we pre-

train RNN modules not on task-specific labels, but on simple synthetic sequences that capture fundamental spatiotemporal symmetries. This process sculpts each module's high-dimensional state space into structured, low-dimensional manifolds–rings, cylinders, and tori–that represent interpretable motion primitives such as direction, velocity, and position. These input-driven manifolds (Sussillo & Barak, 2013; Gallego et al., 2017) exhibit smooth and flexible dynamics: they remain stable under noise and perturbations while adapting rapidly to changing inputs. This enables robust encoding of continuous task variables under strong distribution shift, preserving performance across tasks.

We validate the framework on the DVS (Dynamic Vision Sensor) gesture benchmark and a custom RGB (Red-Green-Blue) rehabilitation dataset. In both domains, the learned manifolds preserve their structures–anchored to interpretable motion variables–even under severe interruptions such as corrupted or missing frames. The pretrained RNN modules can be flexibly composed within a modular architecture, each specializing in distinct spatiotemporal features. A Hebbian inspired mechanism allows lightweight reconfiguration at deployment, requiring only a small adapter to be trained. This enables rapid few-shot adaptation in a single epoch without disrupting the pretrained attractor dynamics. Our framework matches or exceeds the performance of leading deep models, e.g., C3D (Convolutional 3D) (Ji et al., 2012) and ViViT (Video Vision Transformer) (Arnab et al., 2021), while using an order of magnitude fewer parameters and dramatically reducing training energy demands. By unifying biologically inspired attractor dynamics with modular composition, this work provides a practical path toward robust, efficient, and continually adaptive information processing.

## 2 RELATED WORK

Deep models, such as two-stream CNNs (Simonyan & Zisserman, 2014) and GNNs (Yan et al., 2018), achieve strong recognition accuracy but require costly optical flow or skeleton extraction, yielding opaque features and require extensive retraining for each task. Reservoir computing (RC) leverages the dynamics of randomly connected RNNs without training recurrent weights (Poole et al., 2016; Li et al., 2023; Yang et al., 2023), but lacks the inductive biases required for robust transfer. Invariant representation learning for domain adaptation, via augmentation (Chen et al., 2023), contractive autoencoders (Rifai et al., 2011), adversarial alignment (Ganin & Lempitsky, 2015; Shi et al., 2022), or causal mechanisms (Rojas-Carulla et al., 2018), is typically data-intensive and produces latent features with limited interpretability. In contrast, our framework introduces **explicit attractor-based invariance** that enables robust transfer with minimal training.

## 3 MODEL

Many human actions involve low-complexity, constrained motions that resemble rigid-body dynamics (Johansson, 1973; An, 1984). Recognizing such actions can often be reduced to identifying a few core spatiotemporal observables, such as position, direction, and velocity of movement (Fig. 1a). We hypothesize that RNNs pretrained to extract these elementary features from synthetic sequences can develop reusable inductive biases that generalize to naturalistic settings. To test this, we pretrain vanilla RNNs on synthetic, task-agnostic video sequences that isolate specific motion attributes (Williams & Zipser, 1989) (Fig. 1b). Each RNN specializes in encoding one of three primitives–*direction*, *velocity*, or *spatial salience*–forming smooth low-dimensional manifolds aligned with physical motion parameters. These representations emerge independently of any downstream task, enabling transfer across domains. We integrate these pretrained modules into a Pretrained Reservoir Group (PRG) framework. At deployment, modules are flexibly composed into task-specific pipelines for action recognition, with mechanisms for input alignment, feature encoding, and lightweight adaptive decoding. This design supports efficient adaptation and strong generalization under domain shifts.

To instantiate each module, we employ a vanilla recurrent neural network (RNN) with 512 neurons, where the hidden state $h(t)$ evolves as a leaky integration of the previous state and current input, followed by ReLU and layer normalization for stability:

$$h(t+1) = \text{LayerNorm}\left[\left(1 - \tfrac{1}{\tau}\right)h(t) + \tfrac{1}{\tau}\,\text{ReLU}\left(W_{\text{rec}}h(t) + W_I I(t)\right)\right], \tag{1}$$

with $\tau = 2$, input weights $W_I$, and recurrent weights $W_{\text{rec}}$.

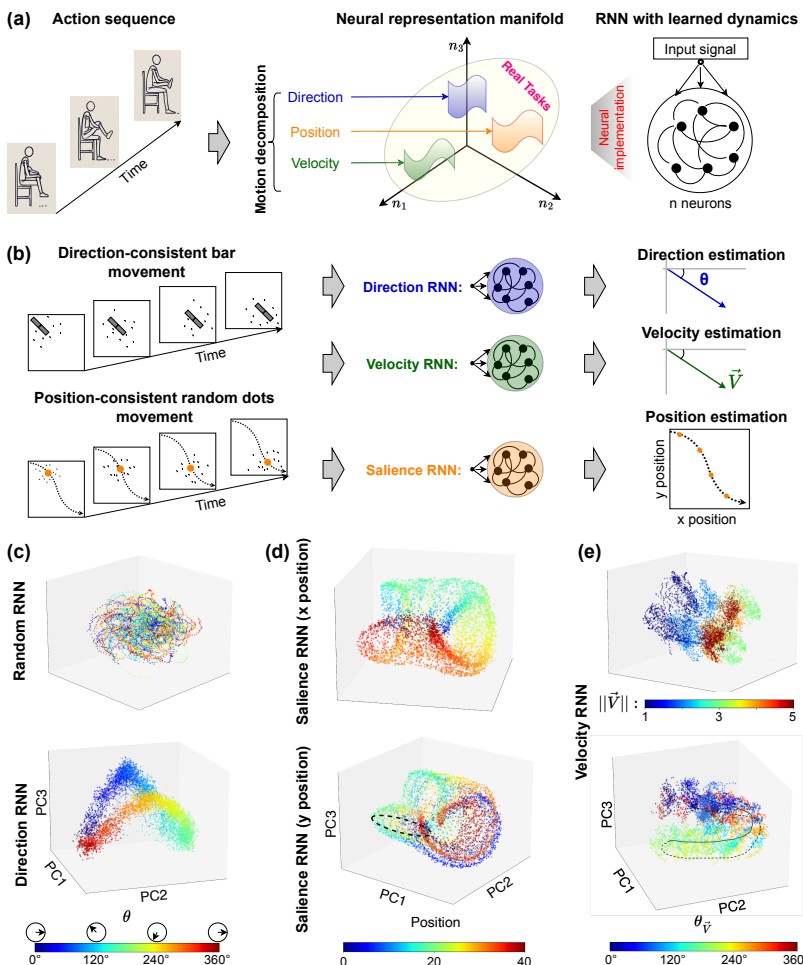

Figure 1: Overall concept, pretraining schema, and emergent neural manifolds. (**a**) Concept: Action sequences are decomposed into core motion attributes–direction, velocity, and position–captured by specialized pretrained RNN modules as low-dimensional manifolds. (**b**) Pretraining tasks: Each module is pretrained on synthetic motion primitives (bars or dots) to encode its target attribute. (**c-e**) Principal component (PC) projections of emergent dynamics. (**c**) Direction RNN: A smooth ring manifold aligned with input direction emerges after pretraining (bottom), contrasting with the unstructured dynamics of a random RNN (top); (**d**) Salience RNN: A toroidal manifold aligned with $x$ (top) and $y$ (bottom) positions; Black dashed line indicates trajectory at a fixed azimuthal angle. (**e**) Velocity RNN: A structured cylindrical manifold jointly encoding motion speed $|\vec{V}|$ (top) and velocity direction $\theta_{\vec{V}}$ (bottom).

Three such RNNs were pretrained from random initialization on synthetic video sequences, each isolating a distinct motion attribute. The **Direction RNN** was trained on $T = 100$-frame bar-motion clips spanning 12 directions, with Bernoulli noise ($p = 0.01$), to classify motion angle $\theta$. The **Velocity RNN** was trained on bars moving at five discrete speeds across 12 directions, jointly predicting direction and speed. The **Salience RNN** was trained on dot-cloud clips with a smoothly drifting Gaussian centroid, estimating $(x, y)$ positions on a discretized grid. This pretraining circumvents the challenges of task-specific end-to-end optimization (e.g., vanishing/exploding gradients (Pascanu et al., 2013)) and induces dynamics aligned with core kinematic primitives (details in Appendix A.1).

These pretrained RNNs are organized in the PRG framework that draws on principles of cortical organization and integrates motion-related attributes effectively to achieve robust, interpretable, and adaptive gesture cognition across tasks. The architecture consists of three components (Fig. 2): an

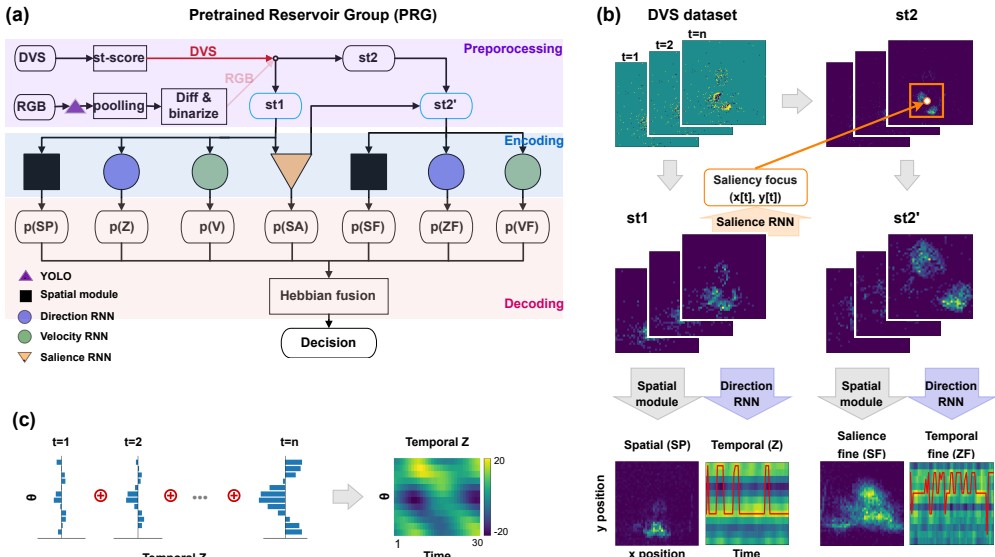

Figure 2: PRG framework. (**a**) Overview of the PRG framework. (**b**) DVS input processing: At each time point $t = 1, ..., n$, preprocessed DVS frames (**st1**) drive Salience RNN to estimate focus points $(x[t], y[t])$, which crop a focused frame (**st2$'$**). Both **st1** and **st2$'$** are then fed into pretrained RNN modules to extract spatial and temporal(direction and velocity) features. (**c**) Temporal encoding in the $Z$ module: Motion directions $\theta$ are binned into histograms over time and concatenated into a time-direction map, capturing trajectory evolution for downstream decoding.

input alignment module, a set of pretrained RNN modules, and a fusion layer for evidence accumulation. The alignment module merges data from diverse sources into a unified binary video format at multiple spatial resolutions, ensuring compatibility with all pretrained modules. Each pretrained RNN module is specialized for a distinct kinematic attribute, and their outputs are combined through a structured, biologically inspired fusion process.

Mirroring cortical parallelism, we design multiple RNNs tuned to complementary motion attributes: 1) A **Spatial Module**, with strong self-connections, integrates pixel-wise activity to capture spatial occupancy; 2) A **Direction RNN** encodes motion angle into 12 angular bins, yielding coarse directional features $(Z_+, Z_-)$; 3) A **Velocity RNN** encodes trajectory speed into 5 discrete levels, producing $(V_+, V_-)$; 4) A **Salience RNN**, inspired by the attentional mechanism in the frontal eye field (FEF) (Schall, 2004; Thompson et al., 2005), predicts motion centroids $(x(t), y(t))$, which define high-resolution patches that are further processed by the Direction and Velocity RNNs, yielding refined estimates $(ZF_+, ZF_-, VF_+, VF_-)$ (more details in A.3).

Finally, decision-making is modeled after parietal–prefrontal circuits that integrate heterogeneous evidence over time (Gold & Shadlen, 2007; Ernst & Banks, 2002). We implement a two-stage fusion: 1) **Static fusion**: each module's hidden state is mapped to class probabilities (via SVM) and uniformly aggregated. 2) **Adaptive Hebbian fusion**: Module fusion weights, applied to the class probabilities, are dynamically updated on a small validation set. This reinforcement of correct associations enables rapid few-shot adaptation without full retraining (see Fig. 10; details in Sec. A.4).

This modular design decomposes complex gestures into interpretable components (A.11), supports robust cross-domain transfer, and provides a principled path toward brain-inspired generalization.

## 4 RESULTS

### 4.1 HIDDEN REPRESENTATIONS OF EACH RNN MODULE

To examine the emergent dynamics, we recorded hidden states from each pretrained RNN under the correspondent synthetic video sequences and projected them into 3D subspaces via principal

component (PC) analysis. Pretraining produced strikingly structured low-dimensional manifolds embedded in high dimensional neural space. 1) **Direction RNN**: hidden states self-organized into a smooth ring manifold (effective dimension $\approx 2.11$, measured by participation ratio A.6), each point encoding a distinct motion angle–reminiscent of biological ring attractors for head direction or angular velocity (Khona & Fiete, 2022) (Fig. 1c). 2) **Salience RNN**: hidden states formed a toroidal manifold (effective dimension 3.81), where positions on the torus mapped to $(x, y)$ spatial coordinates, embedding 2D locations into a continuous latent structure (Fig. 1d). 3) **Velocity RNN**: hidden states traced a cylindrical manifold (effective dimension 3.46), with angular components encoding motion direction and radial or axial coordinates capturing speed (Fig. 1e). In all cases, the relevant motion feature emerged as an *order parameter*, around which activity self-organized into a smooth, low-dimensional manifold. Thus, synthetic attribute-specific pretraining can reliably sculpt modular RNNs into interpretable dynamics.

### 4.2 PROBING HIDDEN DYNAMICS

We analyze the internal dynamics of each RNN under perturbations of both internal states and external inputs (Fig. 3a, b, and for methods refer to A.5). Across modules, emergent manifolds are low-dimensional, anchored to inputs, and resilient to both internal noise and external input variation. The dynamics are fully *input-driven*: internal states closely follow current stimuli rather than sustaining themselves autonomously. For example, in the **Direction RNN**, a directional input (e.g., $I_\theta = 90°$) positions the state on the ring manifold; removing the input rapidly drives activity back to baseline. Unlike autonomous attractors, these modules function as input-driven continuous attractors tailored to fast-changing streams. The manifold consists of input-dependent fixed points with two core dynamical properties (Fig. 3c-e): 1) **Stability**: Perturbations of internal states under a fixed input contract back to the manifold within $5 \sim 10$ steps. Vector fields show strong restorative flows toward input-anchored regions, with restoring forces increasing with distance—consistent with input-driven fixed points. 2) **Adaptability**: Small input shifts (e.g., motion angle $\pm 30°$, position $\pm 1$, or velocity $\pm 1$) induce smooth tangential flows toward the fixed point corresponding to the new input. A high cosine similarity between the flow and the manifold's tangent directions confirms that the geometry is preserved, thereby enabling stable and continuous tracking.

These properties drive strong out-of-distribution generalization: 1) **Direction RNN** maintains over $80\%$ decoding accuracy under novel combinations of direction change frequency and speed (Fig. 3f), far outperforming a C3D (Ji et al., 2012) baseline, which exhibits $> 40\%$ error at high velocities; 2) **Salience RNN** generalizes across centroid moving speed and spatial spread $\sigma$, again surpassing C3D (Fig. 3g); 3) **Velocity RNN** retains high accuracy under unseen velocity-change frequencies and speed ranges, demonstrating strong resilience to distribution shifts (Fig. 3h). Overall, synthetic pretraining sculpts rings, tori, and cylinders–smooth, input-driven manifolds that are dynamically attractive and geometrically stable (Sussillo & Barak, 2013). These manifolds act as strong inductive biases, supporting robust extrapolation beyond training distributions.

### 4.3 REPRESENTATION TRANSFER TO REAL-WORLD TASKS FOR DOMAIN ADAPTATION

Crucially, the pretrained manifolds (**input-driven continuous attractors**) transfer directly to real-world tasks, preserving both geometric structure and dynamical properties. This supports our core claim: low-dimensional attractor manifolds act as robust, reusable computational substrates for naturalistic motion recognition without retraining.

We validated this on two real-world datasets: the DVS Gesture benchmark and a custom RGB rehabilitation action dataset. For each, we projected real-task neural activity into the same PC spaces defined during synthetic pretraining, directly testing whether trajectories remained confined to the original manifolds. Despite the sharp distribution shift from synthetic videos (direction, salience, velocity) to real-world tasks (Fig. 6), both topology and flow dynamics persisted (Fig. 4). Two key transfer conditions emerged. 1) **Dynamical confinement:** Real input streams induce smooth, low-dimensional flows that remain near the pretrained manifold's tangent space, ensuring stable encoding. 2) **Geometric alignment:** The pretrained manifold structure provides a topological metric for real task inputs, embedding relevant motion features–direction, spatial position, and velocity–into continuous latent coordinates.

These principles generalize across all modules (Fig. 4). 1) **Direction RNN**: Clockwise (CW) and counterclockwise (CCW) arm-waving gestures trace oppositely circulating trajectories around the

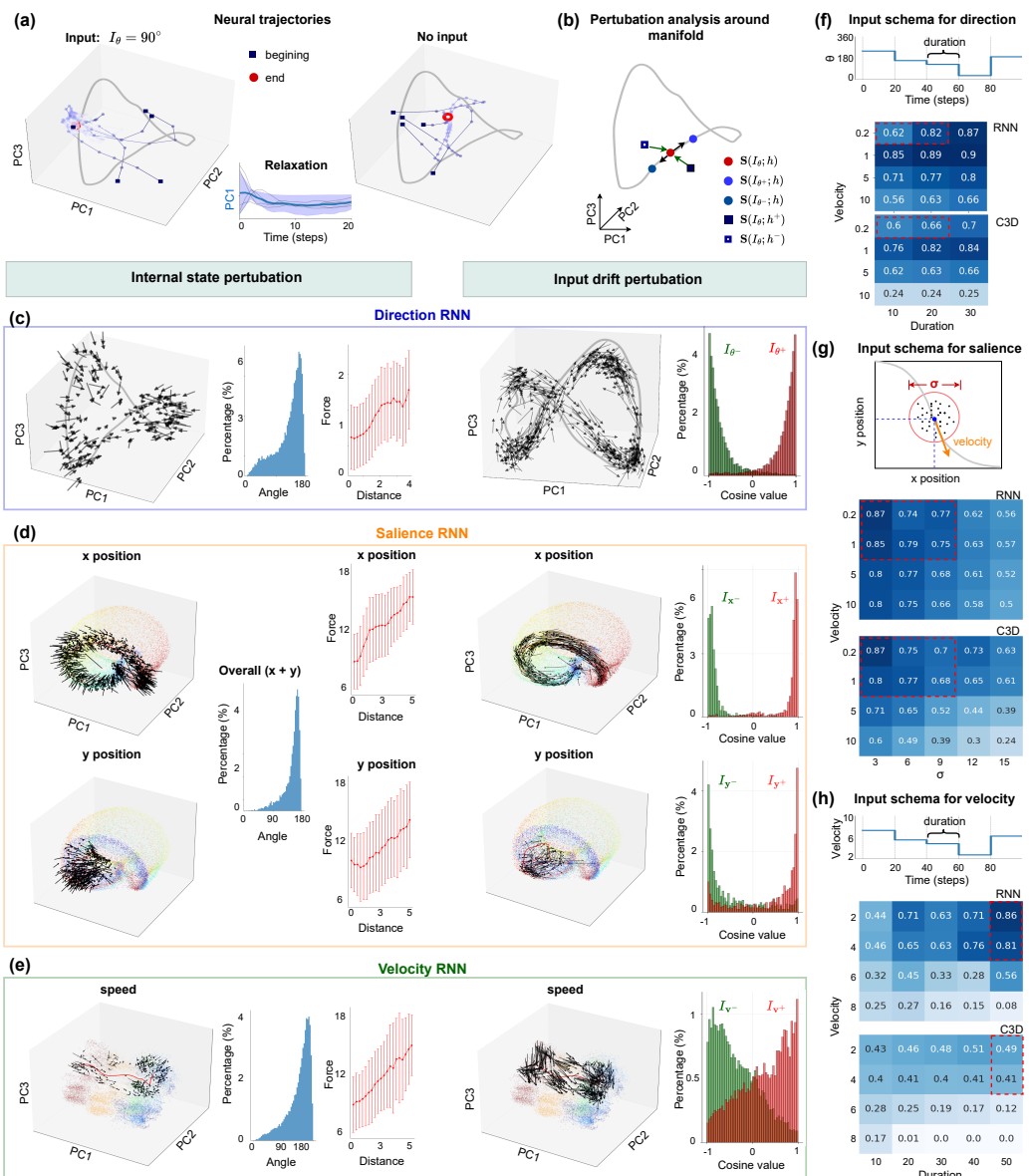

Figure 3: Neural dynamics around attractor manifolds in direction (**c**), salience (**d**), and velocity (**e**) RNNs. (**a**) Input-driven dynamics: Activity converges to a stable location on the ring manifold under a fixed input ($I_\theta = 90°$) and decays when input is removed. (**b**) Perturbation types: Internal perturbations (off-manifold) vs. input shifts (along-manifold). (**c**) Direction RNN: Internal stability analyses (vector fields, force–distance profiles, angle histograms) show recovery toward the manifold using manifold analysis (A.5). Input shift analyses (vector fields, tangent-angle statistics) reveal smooth adaptation under opposite offsets. (**d**, **e**) Same analyses for Salience and Velocity RNNs, respectively; point colors in PC spaces match Fig. 1. (**f**-**h**) Generalization performance across input direction, duration, and velocity variations, compared to a C3D baseline trained under the same paradigm. Red squares denote training conditions.

ring manifold (Fig. 4a, left). Trajectories align with tangent flow (cosine similarity; Fig. 4a, middle), yielding 97% classification accuracy and robustness to frame dropout (random deletion of a proportion of frames) and interpolation (insertion of random clips between consecutive frames A.7), surpassing C3D (Ji et al., 2012) (Fig. 4a, right). 2) **Salience RNN**: Left- vs. right-hand waves map to spatially distinct toroidal regions (Fig. 4b, left). Decoded $x$-positions show clear separation (Fig. 4b,

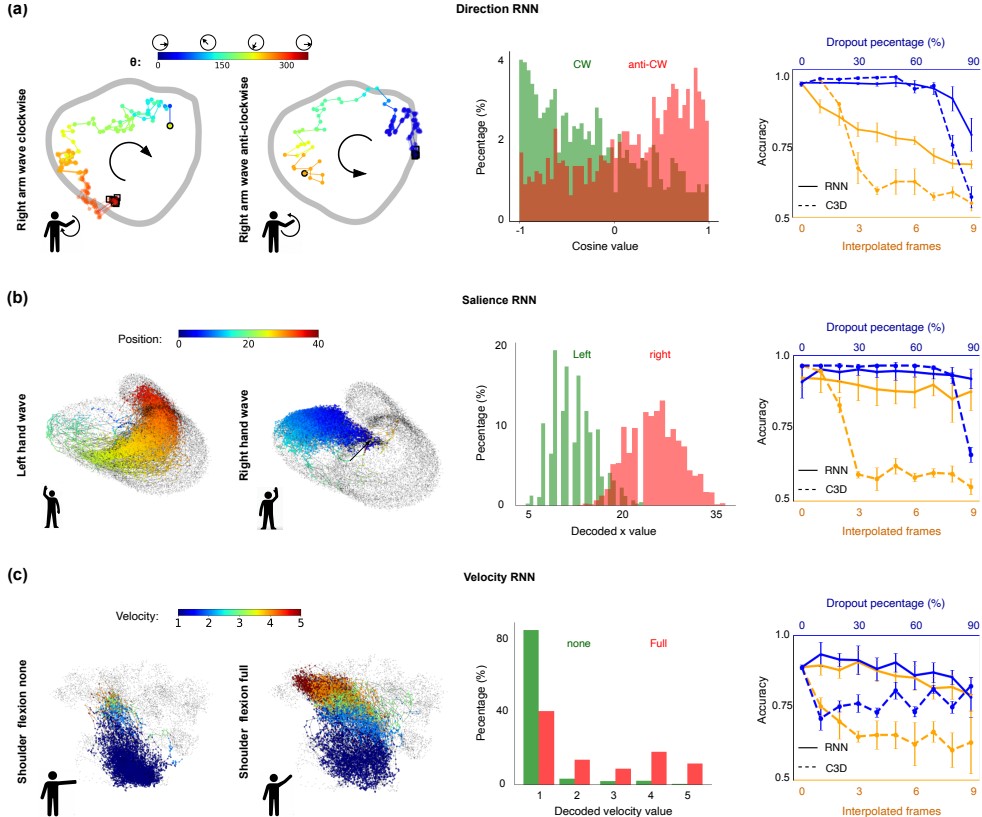

Figure 4: Task-specific neural dynamics and robustness of pretrained RNNs in real motion tasks. (**a**) Direction RNN ($\theta$): Clockwise and counterclockwise arm-waving gestures evoke opposite trajectories on the ring manifold (left); Cosine similarity confirms flow alignment with tangent vectors (middle); Decoding remains robust under frame perturbations, outperforming C3D (right). The x-axis indicates the proportion of frames deleted or random clips inserted between consecutive frames. (**b**) Salience RNN (position $x$): Left vs. right hand waves map to spatially distinct regions on the toroidal manifold (left); Decoded $x$-positions show clear separation (middle); Decoding accuracy surpasses C3D (right). (**c**) Velocity RNN (speed $|\vec{V}|$): Shoulder movements of different intensities map to distinct cylindrical trajectories (left); Decoded speed levels capture motion intensities (middle); Decoding accuracy remains stable under perturbations, exceeding C3D (right).

middle), preserving the spatial structure learned in pretraining under naturalistic inputs. Robustness tests confirm stable decoding under noisy inputs (Fig. 4b, right). 3) **Velocity RNN**: Shoulder movements of different intensities trace distinct cylindrical trajectories (Fig. 4c, left). Decoded speed levels reliably capture motion strength, including null and full movements in the RGB rehabilitation actions (Fig. 4c, middle), and remain robust under temporal perturbations (Fig. 4c, right).

**Most critically, we find that manifolds violating the low-dimensional property exhibit dramatically reduced transferability.** When additional features are entangled during pretraining, the corresponding manifold becomes mixed and its dimensionality increases (details in A.8). As a consequence, both geometric alignment and dynamical stability break down during transfer, resulting in poor task performance (Fig. 7). In contrast, low-dimensional attractors that encode disentangled motion attributes preserve topological invariance, ensure smooth flow alignment, and maintain robust performance under noise and distributional shifts.

Together, these results demonstrate that **low-dimensional, input-driven continuous attractors** are not only biologically plausible but also necessary for reliable and generalizable representation transfer. They provide a principled mechanism for robust generalization–one that static feedforward models like C3D fail to match, especially under nonstationary or corrupted conditions (Fig. 8).

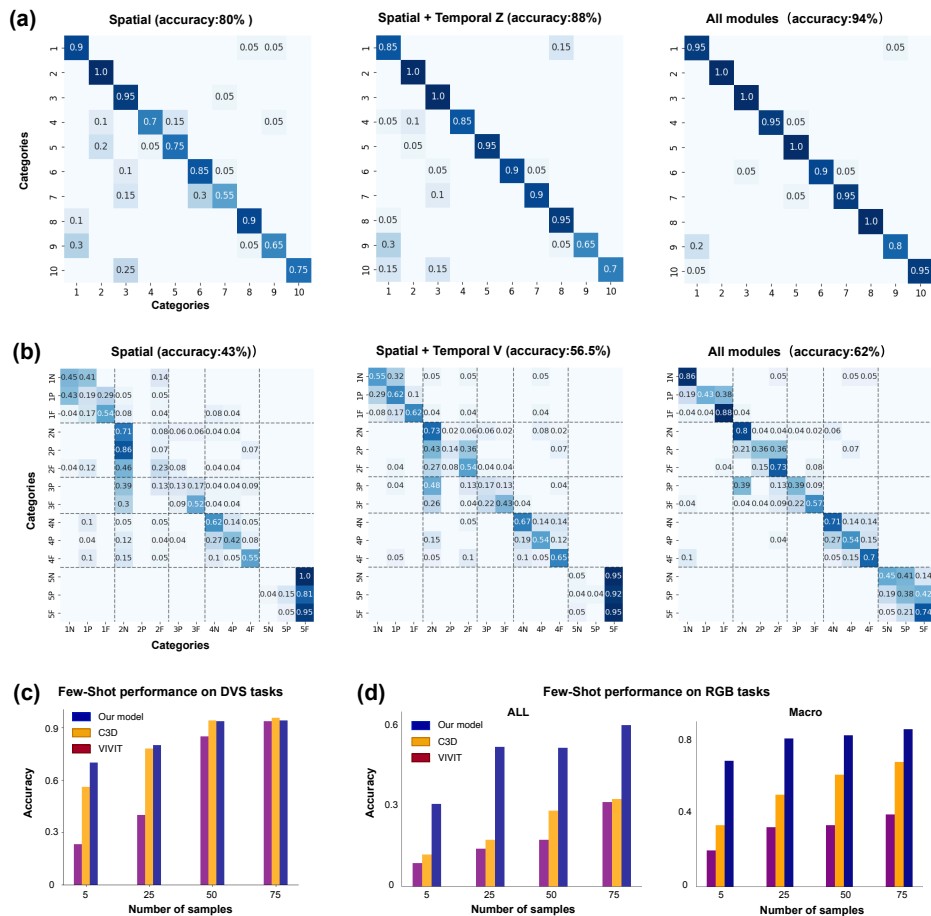

Figure 5: Model performance on DVS and RGB tasks. (**a**) DVS-task Confusion matrices: Classification accuracy using spatial module only (left), spatial+temporal (direction) modules (middle), and all modules (right). Darker colors indicate higher accuracy. (**b**) RGB-task Confusion matrices: Same comparison across module combinations. (**c**) DVS-task few-shots learning performance comparison. (**d**) RGB-task few-shots learning performance comparison: Overall (left) and macro-averaged (right) accuracy across different sample sizes.

## 4.4 PERFORMANCE AND ABLATION STUDIES ACROSS TWO DATASETS

We evaluated the full PRG system on two datasets: the DVS gesture dataset and a custom RGB-based rehabilitation action dataset. The rehabilitation dataset comprises 14 fine-grained action classes involving upper- and lower-limb joints (e.g., shoulder, elbow, and knee), designed to simulate realistic rehabilitation scenarios with diverse camera angles and subject variability (more details in A.2). Our compact model with $\sim$ 5-million (5M) parameters achieves high accuracy, interpretability, and robustness, even in data-limited settings. Despite its small size, it outperforms substantially larger baselines, including ResNet C3D (10- and 50-layer) and ViViT, which require orders of magnitude more parameters ($240 \sim 300$M) and far longer training times (hours vs. minutes).

On the DVS dataset, our model reaches $94\%$ accuracy across 10 gesture classes. Ablation highlights the contribution of each module: 1) **Spatial Module** alone achieves $80\%$ accuracy by capturing location cues (e.g., left vs. right hand waves); 2) Adding **Direction RNN** (Temporal $Z$) boosts performance to $88\%$ by encoding coarse trajectory directionality (e.g., CW vs. CCW rotations in classes 4&5); 3) Incorporating **Salience RNN** and fine-grained features further improves accuracy to $94\%$, enabling the system to resolve subtle gesture variations (e.g., amplitude differences in classes 1&9). On the RGB rehabilitation action dataset, our full model achieves $90\%$ macro-class accuracy

Table 1: Performance on the DVS gesture dataset: all 10 classes, classes 4&5 and 1&9.

| Network | All (10) | 4 vs. 5 | 1 vs. 9 |
|---|---|---|---|
| End-to-End trained RNN | 0.46 | 0.50 | 0.55 |
| C3D(10) | **0.94** | **0.98** | 0.81 |
| ViViT | 0.93 | 0.87 | 0.51 |
| Temporal $Z$ | 0.66 | 0.90 | 0.70 |
| Temporal $Z$ + Spatial | 0.88 | 0.95 | 0.83 |
| Our model: Temporal $Z$+ Spatial + Salience | **0.94** | **0.98** | **0.85** |

Table 2: Performance on the rehabilitation RGB dataset.

| Action type | Our model | C3D(10) | C3D(50) | ViViT |
|---|---|---|---|---|
| Macro class | **0.90** | **0.90** | 0.86 | 0.75 |
| Action 1 | **0.81** | 0.58 | 0.56 | 0.44 |
| Action 2 | **0.81** | 0.55 | 0.49 | 0.50 |
| Action 3 | **0.78** | 0.40 | 0.30 | 0.41 |
| Action 4 | **0.63** | 0.32 | 0.32 | 0.35 |
| Action 5 | **0.49** | 0.44 | 0.41 | 0.43 |
| overall (14 classes) | **0.62** | 0.46 | 0.43 | 0.36 |

and 62% overall subclass accuracy across all 14 categories, substantially outperforming both the ResNet C3D (10- and 50-layer versions) and ViViT baselines. The ablation results show: 1) **Spatial Module** alone achieves 43% overall accuracy with 79% macro-class accuracy and 55% averaged subclass accuracy; 2) Adding **Velocity RNN** (Temporal $V$) improves performance to 56.5% overall with 85% macro-class and 66% subclass accuracy; 3) Incorporating **Salience RNN** and fine-detail refinements raises performance to 62% overall with 90% macro-class and 71% subclass accuracy. Remaining fine-grained misclassifications (e.g., elbow vs. knee motions) largely stem from the inherent difficulty of projecting 3D joint dynamics onto 2D image under diverse viewpoints and distances. Nonetheless, the system robustly distinguishes coarse categories (e.g., shoulder vs. elbow motions), making it practical for physiotherapy monitoring and other real-world applications.

In few-shot scenarios, our system demonstrates strong sample efficiency: 1) On **DVS dataset**, with only 5 samples per class, it achieves $\approx 70\%$ accuracy, far exceeding 3D ResNet-50 (58%) and ViViT (28%); 2) On **RGB dataset**, with 5 samples per class, it attains $\approx 75\%$ macro-class accuracy, compared to $\approx 20\%$ for ResNet C3D and ViViT. Accuracy continues to improve steadily with additional samples, reaching nearly 80% macro-class accuracy for 25 samples per category. These results highlight the benefits of modular pretraining and Hebbian-style adaptive fusion, enabling low-resource gesture recognition with interpretability and adaptability in real-world rehabilitation and monitoring contexts.

## 5 CONCLUSION

In summary, our framework combines the stability of classical attractor networks with the adaptability of input-driven RNNs through task-agnostic pretraining. We sculpt invariant input-driven continuous attractors that maintain stable, low-dimensional representations even under abrupt input changes, enabling robust representation transfer and few-shot adaptation in nonstationary, real-world settings. This biologically inspired modular design also allows agents to expand their attractor repertoire through synthetic pretraining–using, for example, generative models or physics-based simulators–and to dynamically select and compose modules via meta-learning or neural architecture search. Together, these capabilities lay the groundwork for resilient, low-cost embodied AI that can learn continually and adapt in real time with minimal data and computation.

## ETHICS STATEMENT

I acknowledge that I and all co-authors of this work have read and commit to adhering to the ICLR Code of Ethics.

## REPRODUCIBILITY STATEMENT

All data needed to evaluate the conclusions in the paper are present in the paper and/or the Supplementary Materials. The custom RGB action dataset is available at `https://doi.org/10.5281/zenodo.16454040` and `https://doi.org/10.5281/zenodo.16473362`. Computer code for all simulations and analysis of the resulting data is available at `https://doi.org/10.5281/zenodo.16441066`.

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

## A    METHOD DETAILS

### A.1    RNN PRETRAINING

All three RNN modules are obtained through a task-agnostic pretraining method with synthesized input videos designed as follows. To ensure versatility across various scenarios, we employ the original vanilla recurrent neural network (RNN) with 512 neurons, with an input weight matrix $W_I$, an internal matrix $W_{\text{rec}}$, and a readout matrix $W_O$. The decay timescale $\tau$ is set to be 2 time steps. The internal matrix $W_{\text{rec}}$ is initially a sparse matrix with $80\%$ positive and $20\%$ negative values, mimicking the E-I network in the cortex (Pastore et al., 2018).

The hidden state $h(t)$ in the RNN evolves as:

$$\tilde{h}(t+1) = \left(1 - \frac{1}{\tau}\right) h(t) + \frac{1}{\tau} \text{ReLU}\left(W_{rec}h(t) + W_I I(t)\right),\tag{2}$$

$$h(t+1) = \frac{\tilde{h}(t+1) - E\left[\tilde{h}(t+1)\right]}{\sqrt{Var\left[\tilde{h}(t+1)\right] + \epsilon}},\tag{3}$$

where $I(t)$ represents the input frame at time $t$ flattened into a 1D vector, $\epsilon$ is a small positive offset, and $E[\cdot]$ ($Var[\cdot]$) denotes the mean (variance) of the signal. To maintain consistency across different input conditions and enhance network stability, layer normalization (Eq. 3) is applied at each time step, ensuring that the hidden state $h(t)$ remains normalized throughout the network's operation.

A linear readout function and cross-entropy loss are used to train the network:

$$Z(t) = W_O h(t) + b,\tag{4}$$

$$P_c(t) = \frac{\exp\left(Z_c(t)\right)}{\sum_{j=1}^{\text{Classes}} \exp\left(Z_j(t)\right)},\tag{5}$$

$$S = -\alpha \sum_{t=1}^{T} \hat{P}(t) \log P(t),\tag{6}$$

where $\hat{P}$ is the one-hot ground truth label, $P$ is the estimated probability, Classes is the number of categories, $T$ is the total time steps per clips. The optimization uses truncated backpropagation through time (Werbos, 1990) with an Adam optimizer at a learning rate of $10^{-4}$. The input video and training target vary across the direction, salience, and velocity networks, and their respective network configurations are provided in Tab. 3. All models converge within $10^6$ training episodes, requiring approximately 48 hours of computation on a single NVIDIA A40 GPU.

### A.2    DATA ACQUISITION AND PREPROCESSING

We evaluate our system using two distinct datasets. The DVS Gesture Dataset (Amir et al., 2017) provides event-based recordings particularly suited for motion recognition tasks. Additionally, we curated a custom RGB-based rehabilitation action dataset inspired by the Fugl-Meyer Assessment (Gladstone et al., 2002), comprising 1,678 samples across five major categories (Fig. 9):

(1) Shoulder flexion from $90°$ to $180°$

(2) Shoulder flexion from $0°$ to $90°$

(3) Shoulder abduction from $0°$ to $90°$

(4) Pronation and supination

(5) Knee flexion to $90°$

Each category is further subdivided into three recovery levels: none (0), partial (1), and full (2), yielding 15 subcategories in total. Since categories (2) and (3) are indistinguishable at level 0 (none), we merge these two cases into a single group, resulting in 14 effective categories.

The gestures are recorded using an RGB camera from various viewing angles and distances to simulate real-world usage scenarios. A summary of the two datasets is provided in Tab. 4.

Table 3: Pretraining configurations and readout functions for recurrent neural networks

| Network | Category | Parameter | Value |
|---|---|---|---|
| Direction RNN | Pretraining | Velocity range
Clip length
Noise model
Prediction target | 0–1(pixel per frame)
100
Bernoulli($p = 0.01$)
Motion direction $\theta$ |
| | Readout | Output variable(s)
Classes
Loss function | $Z(t)$
12
$S$(cross entropy) |
| Salience RNN | Pretraining | Velocity range
Clip length
Noise model
Prediction target | 0–1
50
Gaussian($\sigma \in [2, 8]$)
Centroid $(x, y)$ |
| | Readout | Output variable(s)
Classes
Loss function | $Z_x(t)$, $Z_y(t)$
42($x$), 42($y$)
$S_x + S_y$ |
| Velocity RNN | Pretraining | Velocity range
Clip length
Noise model
Prediction target | 1–5
50
Bernoulli($p = 0.01$)
Speed, direction |
| | Readout | Output variable(s)
Classes
Loss function | $Z_v(t)$, $Z_\theta(t)$
5($v$), 12($\theta$)
$S_v + S_\theta$ |
| Position direction RNN | Pretraining | Velocity range
Clip length
Noise model
Prediction target | 0–1
50
Gaussian($\sigma \in [2, 8]$)
Centroid $(x, y)$, direction |
| | Readout | Output variable(s)
Classes
Loss function | $Z_x(t)$, $Z_y(t)$, $Z_\theta(t)$
(42($x$), 42($y$)), 12($\theta$)
$S_x + S_y + S_\theta$ |

Table 4: Quantitative summary of datasets

| Dataset | Total samples | Total classes | Samples per class | Train/Test split |
|---|---|---|---|---|
| DVS | 1,220 | 10 | 122 | 4:1 |
| RGB | 1,678 | 15 (5 macro×3 levels) | 112 | 4:1 |

To ensure compatibility with both DVS and RGB streams, we design a unified preprocessing pipeline that (1) converts inputs into a standardized binary video format and (2) denoises in both spatial and temporal domains. For DVS data, we employ a dedicated spatio-temporal core (ST-core) structure (Zhao et al., 2022) for noise reduction (Fig. 2b). The ST-core consists of binary neural networks that perform spatial and temporal processing, ensuring effective denoising while preserving critical spatio-temporal information. The spatial computation integrates the data as follows:

$$s_j(t) = H \left( \sum_{i=j_1}^{j_m} d_i(t) - \theta_s \right), \tag{7}$$

where $d_i(t)$ is the value of the $i^{\text{th}}$ pixel at time $t$, $s_j(t)$ is the output pixel, and integration occurs over a detection range of square region $m = \Delta ST_s{}^2$(correspond to $j_1$ to $j_m$ pixel in input frame). The Heaviside step function $H(S - \theta_s)$ equals 1 if the sum $S$ exceeds the spatial threshold $\theta_s$, and

0 otherwise. Temporal computation is performed as:

$$st_j(t) = H\left(\int_{t-\Delta ST_t}^{t} s_j(t')dt' - \theta_t\right), \tag{8}$$

where $\Delta ST_t$ is the temporal integration window and $\theta_t$ is the temporal threshold. The parameters $\Delta ST_s, \theta_s, \Delta ST_t, \theta_t$ control the spatial and temporal receptive fields and thresholds of the ST-core. We generate two binary video resolutions: **st1** ($42 \times 42$) with parameters $3, 2, 3, 2$ and **st2** ($128 \times 128$) with parameters $1, 1, 2, 2$.

For RGB video preprocessing, we first employ YOLOv8 (Redmon et al., 2016) to detect the subject and extract a $500 \times 500$ region centered on the detected bounding box from the original $1920 \times 1080$ video. To generate multiscale inputs consistent with the DVS modality, we apply average pooling with window sizes of $10 \times 10$ and $4 \times 4$, producing two spatial resolutions referred to as **st1** and **st2**.

To capture motion dynamics and align with the sparsity characteristics of DVS input, we compute the pixel-wise temporal difference between each frame and the final frame of the clip. The resulting difference maps are then binarized using a fixed threshold (95%), yielding binary video representations that mimic the temporal sparsity of event-based data.

## A.3 FEATURE ENCODING

Our modular, pretrained RNNs extract 11 complementary feature streams from each input sequence, encompassing spatial layout, motion dynamics, and salience trajectories (Fig. 2, 9).

**Spatial Distribution.** We first collapse temporal dynamics to capture the overall spatial layout. For each pixel $j$, we sum the low-resolution binary frames **st1**$(t)$ over time:

$$SP = \sum_{t=1}^{n} \mathbf{st1}(t), \tag{9}$$

yielding a 2D activity map that encodes zero-order spatial features, reflecting the cumulative spatial distribution of events (Fig. 9).

**Bidirectional Motion Code.** To capture bidirectional motion patterns, we feed both the original and time-reversed low-resolution sequences into pretrained Direction and Velocity RNNs, yielding readout vectors for direction $[z(t), z_-(t)]$ and velocity $[v(t), v_-(t)]$. Concatenating across frames produces global motion codes (Fig. 9):

$$Z_+ = z(1) \,\|\, \cdots \,\|\, z(T), \qquad\qquad Z_- = z_-(1) \,\|\, \cdots \,\|\, z_-(T), \tag{10}$$
$$V_+ = v(1) \,\|\, \cdots \,\|\, v(T), \qquad\qquad V_- = v_-(1) \,\|\, \cdots \,\|\, v_-(T). \tag{11}$$

This bidirectional temporal code summarizes the evolution of motion direction and speed across time, providing a compact representation of global motion dynamics.

**Salience Trajectory.** The Salience RNN tracks the spatiotemporal focus of motion over time, outputting probability distributions $z_x(t)$ and $z_y(t)$ over horizontal and vertical positions, respectively. These are concatenated into per-frame vectors $f(t) = z_x(t), \|, z_y(t)$, which are then sequentially aggregated to form a trajectory (Fig. 9):

$$SA = f(1) \,\|\, f(2) \,\|\, \cdots \,\|\, f(T), \tag{12}$$

resulting in a high-dimensional trajectory that captures attentional shifts throughout the sequence.

**Fine-Detail Encoding** To capture subtle local variations, we leverage high-resolution input frames **st2**. Using the predicted salience center $(x_1^c, y_1^c)$ from the low-resolution stream **st1**, we scale the coordinates to the **st2** resolution as:, $(x_2^c, y_2^c) = (x_1^c, y_1^c) \times \frac{L_{\mathbf{st2}}}{L_{\mathbf{st1}}}$, and extract a centered crop from **st2**, denoted as **st2**$'$. We then apply the same RNN-based feature extractors to **st2**$'$ to obtain fine-grained spatial features $(SF)$, as well as refined direction and velocity codes $(ZF_+, ZF_-, VF_+, VF_-)$. These features are particularly effective for disambiguating gestures that share similar global patterns but differ in localized movements.

Collectively, the eleven feature descriptors $\{SP, Z_+, Z_-, V_+, V_-, SA, SF, ZF_+, ZF_-, VF_+, VF_-\}$ constitute a compact yet expressive representation for general-purpose action encoding.

### A.4 FEATURE DECODING

The decoding pipeline comprises two sequential stages: per-module probabilistic mapping and modular fusion. This design combines efficient SVM-based classification with a biologically inspired evidence integration strategy (Fig. 10).

#### A.4.1 STAGE I: PER-MODULE PROBABILISTIC MAPPING

For each descriptor, a Support Vector Machine (SVM) with a Radial Basis Function (RBF) kernel is trained to produce a class probability vector, $p(X) = \text{SVM}_{\text{RBF}}(X)$, where $X$ denotes the feature descriptor. The decision function is $f_k(X) = \sum_{i=1}^{N} \alpha_{ik} y_{ik} K(X_i, X) + b_k$, with the RBF kernel defined as $K(X_i, X_j) = \exp\left(-\frac{\|X_i - X_j\|^2}{2\sigma^2}\right)$.

We use Platt scaling (Böken, 2021) to convert these decision scores into calibrated probability estimates. The resulting probability that sample $i$ belongs to class $k$ is computed using the softmax function over the scaled decision scores:

$$P(y_{ik} = 1 | X_i) = \frac{\exp(A_k f_k(X_i) + B_k)}{\sum_{l=1}^{C} \exp(A_l f_l(X_i) + B_l)}, \tag{13}$$

where the scaling parameters $A_k$ and $B_k$ are determined by maximizing the log-likelihood function on a calibration set:

$$L(A_k, B_k) = \sum_{i=1}^{N} \sum_{k=1}^{C} \hat{y}_{ik} \log\big(P(y_{ik} = 1 | X_i)\big), \tag{14}$$

where $\hat{y}_{ik}$ denotes the one-hot encoded ground-truth label for sample $i$ and class $k$.

#### A.4.2 STAGE II: MODULAR FUSION

For each task, we perform *module selection* by retaining the top-$k$ modules based on validation accuracy (20% of the training data). The outputs of the selected modules are then fused using either simple multiplicative fusion (static fusion) or adaptive Hebbian learning-based fusion (Fig. 10), with the latter improving few-shot performance (Table 5).

**Static Fusion** Assuming equal reliability, we compute the element-wise product of all module probability vectors(Fig. 10a):

$$p_{\text{simple}} = \bigodot_{j=1}^{K} p_j, \tag{15}$$

where $p_j$ is the probability vector from module $j$.

**Adaptive Hebbian Fusion** We convert each module's probability output $p_j$ to a one-hot activation $B_j$ via winner-take-all and feed these into a Hebbian-learning layer (Do, 1949) with weight matrix $W^H$(Fig. 10b). The weights are updated during training:

$$\Delta W_{(jk),l}^H = \epsilon \hat{Y}_l^i B_{jk}^i - \lambda W_{(jk),l}^H,$$

where $\hat{Y}_l^i$ is the true class indicator, $\epsilon$ is the learning rate(taken as $10^{-3}$), and $\lambda$ is the decay rate(taken as $10^{-5}$). At inference, the final weighted log-likelihood for class $l$ is computed as:

$$S_l = \sum_{j=1}^{k} \sum_{m=1}^{C} W_{(jm),l}^H \log p_{j,m}, \quad \text{and} \quad \hat{y} = \arg\max_l S_l.$$

### A.5 MANIFOLD ANALYSIS

We treat the recurrent neural network (RNN) as a nonlinear dynamical system whose hidden states settle near "slow points" after training (Sussillo & Barak, 2013). Our procedure for identifying these slow-manifold regions is summarized below.

**Recording internal states.** Each pretrained RNN is driven by two stimulus types: (i) *synthetic sequences*: 500 clips of 100 frames covering the full range of motion parameters for manifold characterization; (ii) *real sequences*: the complete DVS gesture set and RGB rehabilitation videos at resolution $\mathbf{st}_1$ for probing representation transfer. For every frame $t$, we store the input $x_t$, its order parameter $\phi_t$, and the hidden state $h(t)$, yielding triplets $\{(x_t, \phi_t, h(t))\}$.

**Perturbation analysis.** We probe stability by controlled deviations of both hidden states and input parameters. For hidden states, we apply a linear perturbation

$$h'_t = h_t + \epsilon \mathbf{v}, \tag{16}$$

where $\epsilon \in \mathbb{R}$ is small and $\mathbf{v}$ is a unit vector along $h_t$. For inputs, order parameters such as direction $\theta$, velocity $v$, or spatial position $(x, y)$ are perturbed,

$$\theta' = \theta \pm \Delta\theta, \; x'_t = x_t(\theta'), \tag{17}$$

$$v' = v \pm \Delta v, \; x'_t = x_t(v'), \tag{18}$$

$$(x', y') = (x \pm \Delta x, \; y \pm \Delta y), \; x'_t = x_t(x', y'). \tag{19}$$

**Relaxation dynamics.** Starting from $h_t$, we iterate the RNN under constant input $x_t$ for $k = 10$ steps:

$$h_t^{(0)} = h_t, \qquad h_t^{(i+1)} = f(h_t^{(i)}, x_t), \; i = 0 \ldots k-1. \tag{20}$$

The relaxed state $h'_t = h_t^{(k)}$ defines the local attractor. The average update $\Delta h_t = (h'_t - h_t)/k$ provides the velocity $v_t = \|\Delta h_t\|_2$ and normalized direction $d_t = \Delta h_t / \|\Delta h_t\|_2$.

**Low-dimensional embedding and visualization.** All recorded hidden states are projected onto the first three principal components of a PCA fit, yielding a 3D representation. States are colored by their order parameter, and the manifold ridge is traced by the centroid of PC coordinates for each discrete parameter value.

**Quantifying contraction.** For each perturbed state we measure the alignment of the update direction with the vector toward the manifold $\hat{p}_t$:

$$\alpha_t = \cos^{-1}(d_t \cdot \hat{p}_t). \tag{21}$$

Angles $\alpha_t$ near $180°$ indicate a restoring force. Plotting $v_t$ against $\mathrm{dist}(h_t, \mathcal{M})$ reveals how contraction speed depends on distance to the manifold.

## A.6    EFFECTIVE DIMENSIONALITY VIA PARTICIPATION RATIO.

To quantify the dimensionality of neural population activity, we adopt the *participation ratio* (PR), a standard measure derived from the eigenvalue spectrum of the covariance matrix of hidden states (Gao et al., 2017). Given eigenvalues $\{\lambda_i\}$, the PR is defined as

$$\mathrm{PR} = \frac{(\sum_i \lambda_i)^2}{\sum_i \lambda_i^2}. \tag{22}$$

PR provides a robust estimate of the *effective dimensionality* of the representation manifold, allowing us to compare how different pretrained modules compress or expand task-relevant dynamics.

## A.7    DECODING TEST OF REPRESENTATION TRANSFER

**RNN–manifold decoding.** Let $\mathbf{h}_t \in \mathbb{R}^H$ be the hidden state of a pretrained RNN at time $t$. We perform PCA on these hidden states (as in the manifold analysis) and retain the top three principal components. Real-task inputs generate a length-$T$ sequence of hidden states $\mathbf{h}_{1:T}$, which are projected onto the top three PCA components to form a 3D trajectory $\mathbf{z}_{1:T}$ with $\mathbf{z}_t \in \mathbb{R}^3$. For decoding, each trial is flattened into a feature vector $\mathbf{x} = \mathrm{vec}([\mathbf{z}_1, \ldots, \mathbf{z}_T]) \in \mathbb{R}^{3T}$, which is then classified by an RBF-kernel SVM to predict the action label. The same pipeline is applied to all datasets (DVS Gesture and the RGB rehabilitation set).

**Robustness to temporal perturbations.** To test decoder stability, we apply two temporal corruptions to the input before projection: (1) *Frame dropout*, which uniformly removes a fixed proportion

*p* of frames; (2) *Random interpolation*, which inserts stochastic spatiotemporal noise. For the latter, each adjacent frame pair is flattened, concatenated, randomly shuffled to destroy spatial locality while preserving pixel-intensity statistics, then reshaped and inserted between the pair. The SVM is trained and evaluated on both clean and perturbed sequences.

### A.8 CONTROLLED REPRESENTATION EXPERIMENT

A natural question is whether transfer performance is preserved when motion attributes (e.g., direction and velocity) are encoded in a more entangled, high-dimensional space. To test this, we designed an alternative pretraining protocol that jointly learns direction and position, yielding a high-dimensional mosaic representation(effective dimensionality 7.7 measured by participation ratio) . The same manifold visualizations and representation analyses were performed. While this network achieves comparable decoding accuracy on the synthetic pretraining task, it exhibits markedly reduced transferability to real action recognition, as shown in Fig. 7.

### A.9 HIERARCHICAL CLUSTERING METHOD

We analyze module representations using the hierarchical clustering method (Murtagh & Contreras, 2012). The processed video **st1** is taken for analysis. These features are transformed by their respective modules into high-dimensional representations. To facilitate analysis, we apply t-SNE to project these representations into a low-dimensional space. We then compute pairwise distances between all data points to form a distance kernel matrix, which serves as input for bottom-up hierarchical clustering. The resulting dendrogram provides a hierarchical structure of the data, allowing us to interpret relationships between instances at different granularities.

### A.10 C3D AND VIVIT NETWORKS

To benchmark our system, we implement two representative video models: a 3D CNN based on ResNet-50 and a Video Vision Transformer (ViViT).

**ResNet-50 C3D.** We inflate a standard ResNet-50 (Tran et al., 2015) to 3D by extending all 2D convolutions. The first layer uses a $7 \times 7 \times 7$ kernel with stride $(1, 2, 2)$, followed by 3D batch normalization, ReLU, and a $3 \times 3 \times 3$ max-pool. We initialize weights with Kaiming initialization and train for 30 epochs on both the DVS gesture and RGB rehabilitation datasets using Adam (initial learning rate $10^{-4}$, halved every 10 epochs; weight decay $10^{-5}$) and a batch size of 16.

**ViViT.** We adopt the Factorised Encoder variant of ViViT (Arnab et al., 2021) to capture spatiotemporal structure. Videos are first partitioned into non-overlapping tublets via a 3D convolution ($2 \times 16 \times 16$ kernel and stride), linearly projected to 1024-dimensional embeddings, and augmented with learnable positional encodings to yield a sequence of 1536 tokens. The transformer encoder contains 12 layers of factorized self-attention, each with pre-norm LayerNorm, 12-head multi-head attention, and a feed-forward MLP with GELU activation. Training uses AdamW (learning rate $5 \times 10^{-5}$, weight decay $10^{-4}$) for 50 epochs with batch size 8 on both datasets.

### A.11 INTERPRETABILITY OF MODULAR REPRESENTATIONS

To verify that the transferred representations remain interpretable in downstream tasks, we conduct fully unsupervised kernel and hierarchical clustering analyses to examine how individual modules encode meaningful features across tasks. These analyses reveal that each module maintains distinctive, reusable representational structures from pretraining through deployment (Fig. 11a). In the DVS task, the **Spatial module** partitions the scene into task-relevant regions (e.g., left/center/right or upper/lower fields), while the **Spatial fine module** further resolves localized and subtle variations–such as distinguishing left-hand from right-hand waves–resulting in compact, well-separated clusters. Meanwhile, the **Direction module** (temporal $Z$) captures global motion patterns, such as CW vs. CCW rotations). These distinctions are evident in the clear cluster boundaries in the dendrograms and the consistent separations in the 2D embeddings (Fig. 11b). In the rehabilitation task, the **Salience**, **Spatial** and **Velocity** (temporal $V$) modules each separate action 1 apart from actions 3, 4, 5, by leveraging different features–such as knee movements captured by the **Spatial module** (Fig. 11c). Additionally, the three temporal modules ($V$, $Z$, and $VF$) characterize subclasses within action 1, cleanly distinguishing low (None) vs. high (Full) movement intensities in the 2D

embeddings (Fig. 11d-f). This clear, modular "alphabet" of motion primitives–absent in monolithic black-box networks–underscores the interpretability and transferability advantage of our framework.

## A.12 DECLARATION OF LLM USAGE

The core method development does not involve LLMs, which are only used for editing, e.g., grammar, spelling, word choice.

# B ADDITIONAL FIGURES AND TABLES

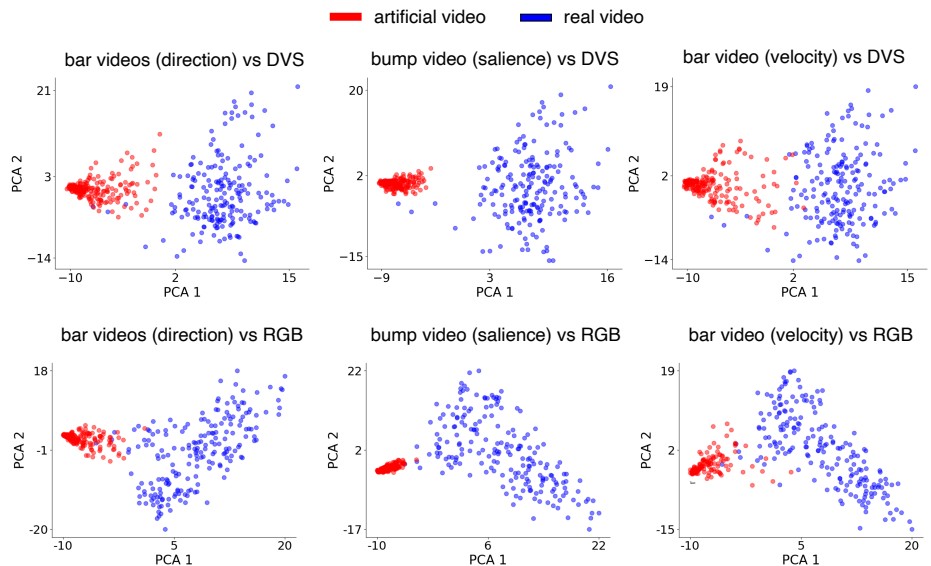

Figure 6: Distributions of artificial pretraining (Direction, Salience, Velocity) and real-task videos (DVS Gesture, RGB Rehabilitation). Each source provides 200 videos, projected into a common PCA space after preprocessing to uniform clip length and flattening.

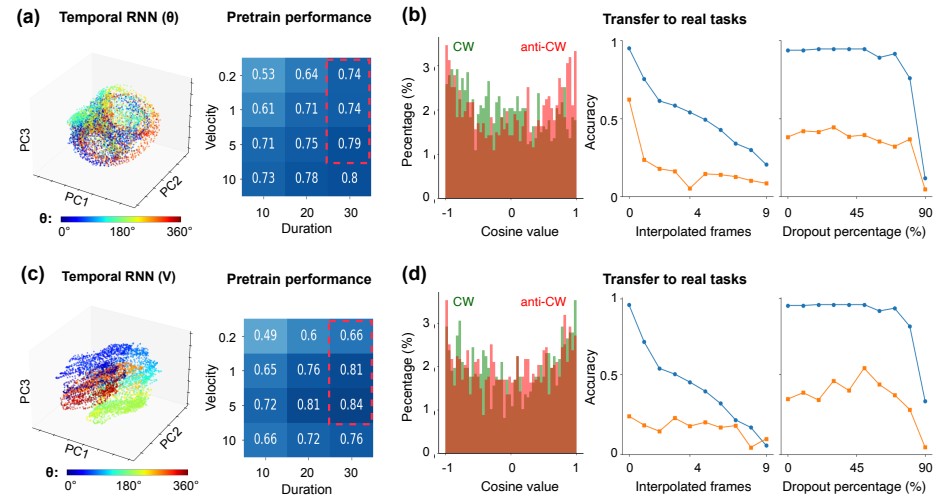

Figure 7: Comparison of RNN hidden-state representations of motion direction in PC space, pretrained on (a,b) position-direction and (c,d) velocity-direction tasks ( A.1). Panels (a,c, left) show structured manifolds of hidden states in the pretrained tasks; (a,c, right) depict pretraining performance across different durations and velocities. Panels (b,d, left) present cosine similarity between flow alignment and manifold tangent vectors in the DVS task for clockwise and counterclockwise motion; (b,d, right) show transfer performance under interpolation and dropout perturbations.

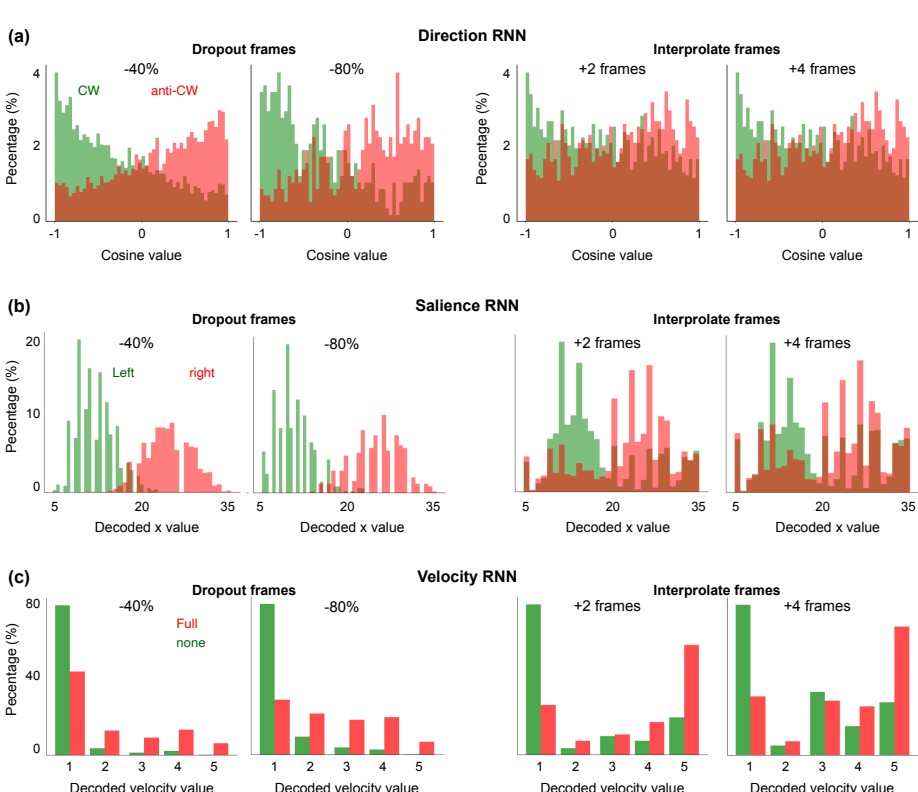

Figure 8: Structural invariance under temporal perturbations. For each RNN module (Direction, Salience, Velocity), we apply the same perturbation protocol: frame dropout (deleting $40\%$ or $80\%$ of frames) and interpolation (inserting 2 or 4 random clips between consecutive frames). (**a**) *Direction RNN (θ):* decoding of clockwise vs. counterclockwise motions from PC states remains accurate under perturbations. (**b**) *Salience RNN (position x):* decoding of left-hand vs. right-hand waves remains robust under perturbations. (**c**) *Velocity RNN (speed $|\vec{V}|$):* decoding accuracy of shoulder-movement levels (full vs. none) remains stable under perturbations. The x-axis indicates the proportion of frames deleted (dropout) or the number of random clips inserted (interpolation).

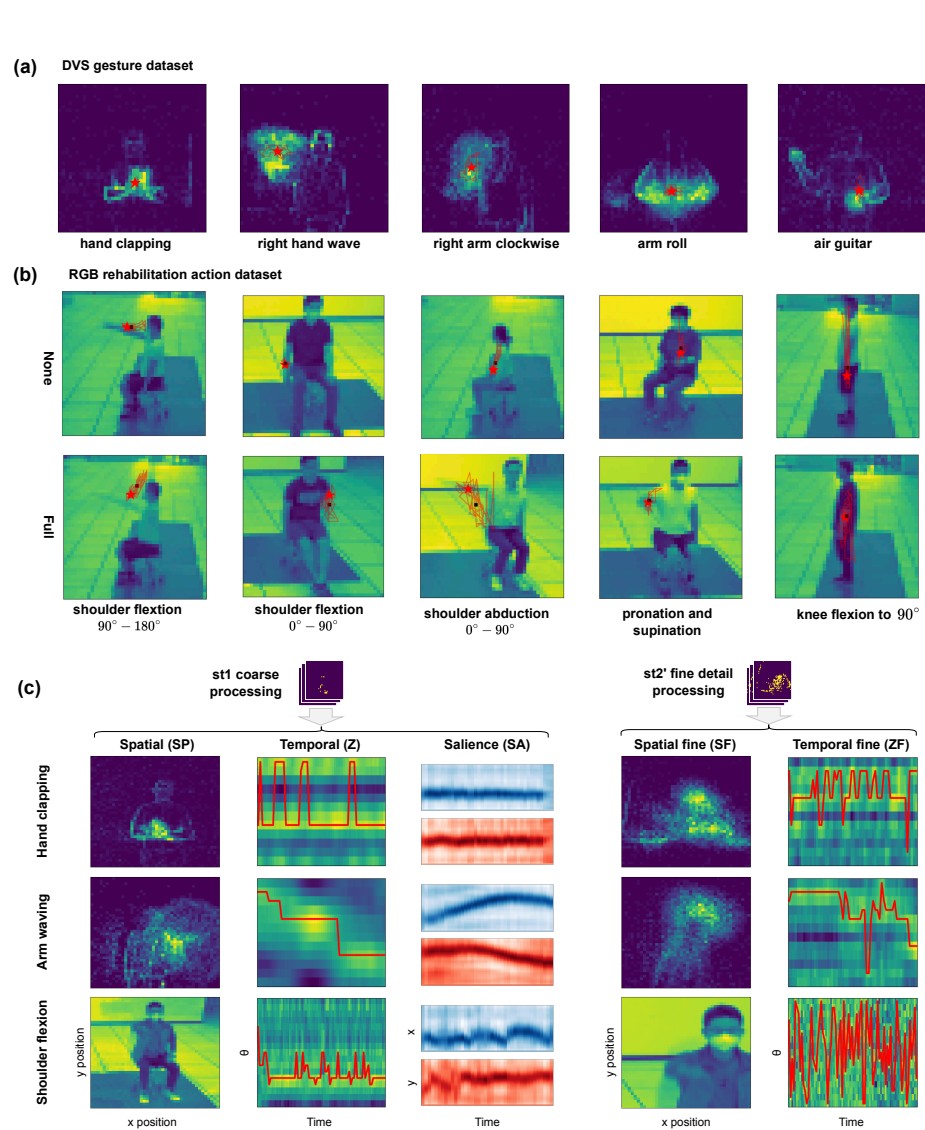

Figure 9: Dataset overview with salience focus position tracking. (**a**) DVS task for different categories with the focus trace provided by salience RNN(red line with the star indicating focus center across frames). (**b**) RGB rehabilitation action task for different categories with the focus trace provided by salience RNN(red line with the star indicating focus center across frames). For the RGB task, five major action categories are illustrated. (**c**) Visualizations of encoding features.

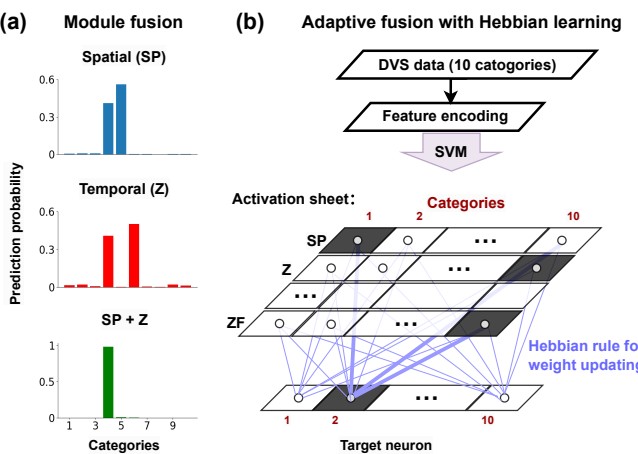

Figure 10: Dataset overview with salience focus position tracking. (**a**) Module predictions: Each RNN module outputs a probability distribution, which is combined via element-wise multiplication to sharpen and improve the final category prediction. (**b**) Hebbian-based adaptive fusion: Module outputs form an "activation sheet" (rows = modules, columns = categories). Hebbian updates (blue arrows) strengthen connections between active modules and the correct output.

Table 5: Performance comparison of Hebbian vs. non-Hebbian fusion across tasks under varying training sample sizes. Columns correspond to the number of labeled training samples per class (5, 10, 20, 50).

| Task / Category | 5 samples | | 10 samples | | 20 samples | | 50 samples | |
|---|---|---|---|---|---|---|---|---|
| | Heb | No-Heb | Heb | No-Heb | Heb | No-Heb | Heb | No-Heb |
| DVS (Gesture) | **0.697** | 0.635 | **0.799** | 0.778 | **0.858** | 0.855 | **0.921** | 0.920 |
| Macro | 0.723 | 0.723 | 0.777 | **0.780** | **0.848** | 0.833 | 0.866 | **0.869** |
| Action 1 | 0.209 | 0.209 | 0.672 | 0.672 | **0.597** | 0.567 | 0.701 | 0.701 |
| Action 2 | **0.551** | 0.539 | **0.449** | 0.416 | 0.685 | 0.685 | **0.652** | 0.640 |
| Action 3 | **0.543** | 0.522 | 0.457 | 0.457 | 0.804 | 0.804 | **0.630** | 0.587 |
| Action 4 | **0.239** | 0.224 | **0.403** | 0.343 | 0.507 | 0.507 | **0.463** | 0.448 |
| Action 5 | 0.299 | **0.313** | **0.403** | 0.343 | 0.328 | 0.328 | **0.522** | 0.507 |

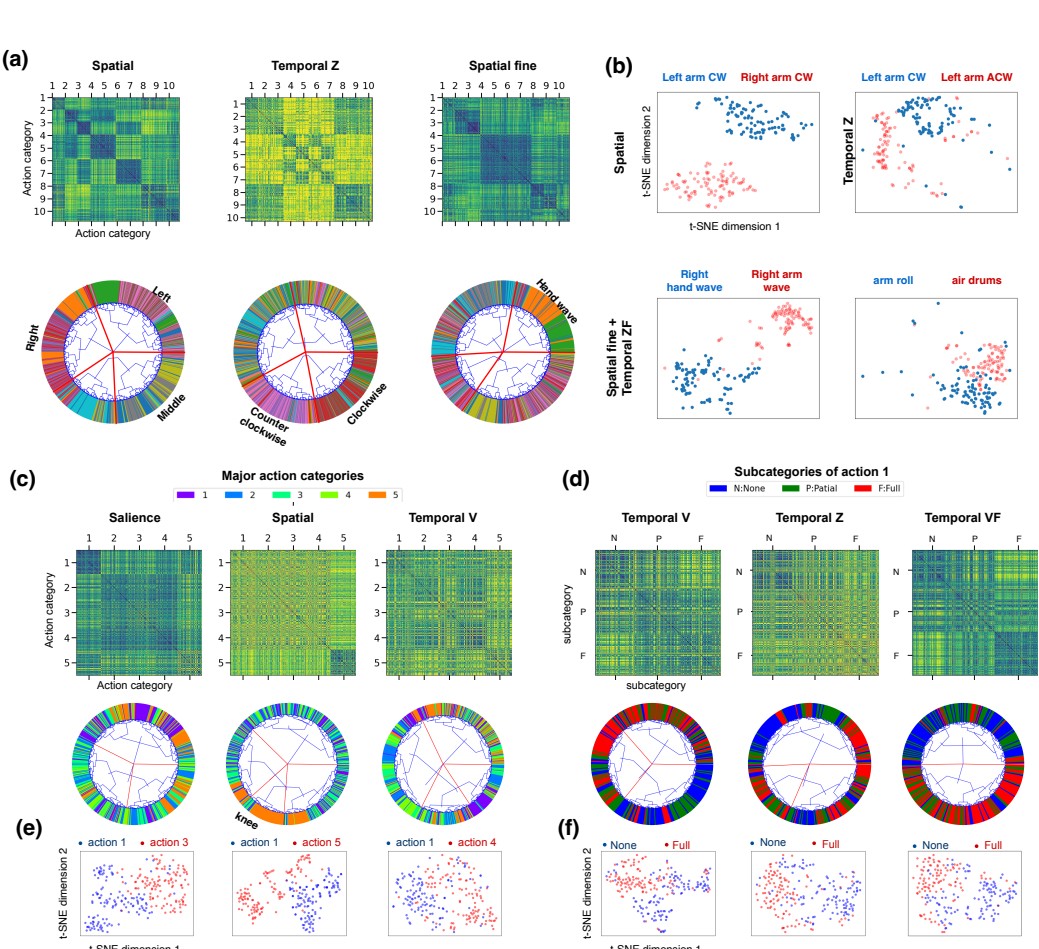

Figure 11: Module-specific Representations across tasks (DVS task: (**a-b**); RGB task: (**c-f**)). (**a**) Kernel matrices (top) show pairwise distances between internal states for each module (spatial (SP), temporal (T), spatial fine (SF)) across 10 action categories, revealing distinct representational patterns; Circular dendrograms (bottom) display how each module clusters actions, capturing structure such as left/right arm movements, motion direction (CW/CCW), and hand-related actions. Cluster labels summarize shared features within groups. (**b**) t-SNE plots show clear separations among actions like "Left arm CW", "Right arm CW", and "Left arm CCW", reflecting strong spatial and temporal selectivity. (**c**) Kernel and clustering analysis across 5 major action categories. (**d**) Kernel and clustering analysis across 3 fine-grained subcategories. (**e**) t-SNE plots showing clear separations among different macro action groups for each module. (**f**) t-SNE plots showing clear separations among different micro action groups for each module.

