# OpenReview forum: "Brain-inspired Representation Transfer through Invariant Input-driven Continuous Attractors in a Modular RNN Framework"
_ICLR.cc/2026/Conference — ICLR 2026 Conference Withdrawn Submission_

### Official Review · Reviewer_MkuA · 2025-10-30

**Soundness:** 2
**Presentation:** 3
**Contribution:** 2
**Rating:** 4
**Confidence:** 2

**Summary:**

This work explores a modular framework composed of recurrent neural network modules pretrained with a simple, task-agnostic procedure to encourage transferable representations. The approach aims to form low-dimensional attractor dynamics that could support stable, task-invariant features. At deployment, only a small adapter layer is trained for new tasks to allow efficient adaptation. Preliminary experiments on the DVS Gesture dataset and a custom rehabilitation action dataset suggest that the method may achieve competitive accuracy with fewer parameters and limited training. These results indicate potential for improving robustness and adaptability in neural systems.

**Strengths:**

- The integration of biologically inspired attractor dynamics with modular neural architectures is a creative and interdisciplinary idea.
- The idea of training only a small adapter at deployment is elegant and computationally efficient, aligning with trends toward parameter-efficient transfer learning.
- The paper is overall well-written, with nice illustrations.

**Weaknesses:**

- The experiments are done only on two benchmarks
- Related work does not seem to be discussed in depth
- The paper mentions “biologically inspired attractor dynamics” and “cortical-like modular composition” without concretely defining how these are implemented or what prior work it builds upon.

**Questions:**

- The related work section is one small paragraph and does not seem to cover the whole literature. How does this paper compare to the related work such as "Warming up recurrent neural networks to maximise reachable multistability greatly improves learning" https://arxiv.org/abs/2106.01001?
- How does the paper connect to "cortical-like modular composition"? What are the references and exact meaning of these terms?

---

### Official Review · Reviewer_SgYC · 2025-10-30

**Soundness:** 2
**Presentation:** 3
**Contribution:** 3
**Rating:** 4
**Confidence:** 3

**Summary:**

The authors proposed a new framework called Pretrained Reservoir Group (PRG), which decomposes movements into three simple modules (direction, velocity, and position) and integrates them through static and Hebbian fusion processes. They used RNNs to pretrain these three movement modules and benchmarked their model against two other models on two datasets.

**Strengths:**

This study is inspired by neuroscience findings that low-dimensional continuous attractors, or neural manifolds, are stable during the same movement tasks. This property could be utilized for transfer learning and is especially suitable in few-shot settings. After decomposing complex movements into three RNN modules, it also reduces the overall parameters in the PRG framework. I believe this idea is original and that this work will have a large influence in both the AI and neuroscience fields.

**Weaknesses:**

The major weakness is the lack of comparison with state-of-the-art methods. The authors compared their model against C3D (Convolutional 3D; Ji et al., 2012) and ViViT (Video Vision Transformer; Arnab et al., 2021). I don’t believe these represent the current SOTA models for video datasets.

**Questions:**

1. Why do the low-dimensional manifolds (ring, cylinder, and torus) correspond to motion primitives such as direction, velocity, and position? Could movement position be represented by a neural manifold in a ring shape?

2. I found that this paper uses either “Position” or “Salience” in its text and figures, which is confusing. What is the difference between them?

3. I think it would be helpful to show the trial-averaged neural dynamics in Figure 1c–e.

4. I don’t understand Figure 3a. In the left panel, four traces start from different locations in the latent space (four blue squares) and converge to the same location (red dot). In contrast, without external inputs, the four traces do not converge at the end. Is my understanding correct?

5. Can all movements or actions be decomposed into three RNN modules?

---

### Official Review · Reviewer_f53W · 2025-11-01

**Soundness:** 2
**Presentation:** 2
**Contribution:** 3
**Rating:** 4
**Confidence:** 3

**Summary:**

The authors pretrain RNNs on synthetic videos, training each seperately to isolate direction, velocity, and spatial salience

These form part of the Pretrained Reservoir Group (PRG) which can flexible learn a wider range of tasks, making use of computational/dynamical motifs learned during pretraining.

The authors go on to benchmark its performance and analyse the activation geometry of the pretrained modules.

**Strengths:**

Well motivated - reuse of recurrent dynamical motifs is an open question in neuroscience. They provide strong empirical evidence of the geometry and dynamics (e.g. stability properties) of their network. The approach is promising (competitive performance with 10x fewer parameters; learning is rapidly transferred across modalities) and their analysis is helpful in understanding the representations developed by modules. Results are presented clearly and intuitively (although presentation weaknesses are discussed below).

**Weaknesses:**

RNNs were trained on separate videos, but for true task agnositicism a stronger argument might have been to train on the same videos with different training objectives/preprocessing. Pretraining tasks that are carefully designed for specific motion primitives (direction, velocity, salience), suggest the approach is actually quite task-specific rather than general-purpose.

Figure 2 was poorly labelled - this made it hard to use for reference while reading the paper.

While the premise is based on flexible adaptation to new tasks, it seems like the PRG is carefully handcrafted to the authors' expectations of where each module is required. This pipeline also depends on an SVM module, which is not biologically motivated, defeating one of the initial motivations of this approach.

Heavily dependent on appendix, very difficult to fully appreciate the approach without reference. e.g. fusion modules were not described at all in main text, and the nature/format of the inputs were also omitted.

Analysis section 4.1 seems to draw fairly intuitive/expected conclusions, namely that the RNN modules were pretrained on low-dimensional tasks display low-dimensional activation manifolds.

**Questions:**

Direction RNN learns a subset of velocity RNN - could this be ablated?

"This pretraining circumvents the challenges of task-specific end-to-end optimization (e.g., vanishing/exploding gradients" - can you explain this? The pretraining is still a supervised learning task, which would suffer from gradient instability.

Have you tried comparing to different pretraining tasks to see how this affects the geometric properties of RNN modules?

---

### Official Review · Reviewer_RTdW · 2025-11-07

**Soundness:** 3
**Presentation:** 3
**Contribution:** 3
**Rating:** 6
**Confidence:** 2

**Summary:**

This paper proposes a predictive coding–based framework for brain-inspired self-supervised learning (SSL). Motivated by neuroscience theories, the authors develop a model that recursively minimizes prediction errors between top-down predictions and bottom-up inputs at each layer, across both time and spatial dimensions. This biologically plausible alternative to backpropagation enables learning without requiring external labels or full-layer error gradients.

**Strengths:**

(1) Training-free: The model avoids traditional backpropagation and relies only on local updates, making it potentially more biologically realistic and hardware-friendly.

(2) The predictive coding mechanism dynamically updates internal representations based on prediction error at each time step.

(3) The model performs robustly across domains, including video understanding and long-tailed recognition, and shows resilience to input corruptions.

**Weaknesses:**

(1) Complexity of policy training: Although framed as “biologically plausible,” the layer-wise update scheme and recurrent inference steps may be computationally expensive and unclear in terms of scaling to deeper architectures or large datasets.

(2) The paper references neuroscience but lacks formal analysis connecting predictive coding to optimal representation learning in modern machine learning terms.

(3) Lack of comparison with state-of-the-art SSL methods (e.g., SimCLR, DINOv2, iBOT)

**Questions:**

(1) How does this model compare to state-of-the-art SSL methods (e.g., SimCLR, DINOv2, iBOT) in high-data regimes?
Could the predictive coding structure be hybridized with modern SSL objectives?

(2) Are there domains (e.g., low-data, highly noisy environments) where the model fails to converge or underperforms compared to baselines?

---

### Note · Authors · 2025-11-24

I have read and agree with the venue's withdrawal policy on behalf of myself and my co-authors.